# Comparison of Two RNA Extraction Methods for the Molecular Detection of SARS-CoV-2 from Nasopharyngeal Swab Samples

**DOI:** 10.3390/diagnostics12071561

**Published:** 2022-06-27

**Authors:** Anna Scarabotto, Simona Balestro, Stella Gagliardi, Rosa Trotti

**Affiliations:** Department of Neurodiagnostics and Services, Laboratory of Clinical Biochemistry (SmeL), IRCCS Mondino Foundation, Mondino 2 st, 27100 Pavia, Italy; anna.scarabotto@mondino.it (A.S.); simona.balestro01@universitadipavia.it (S.B.); rosa.trotti@mondino.it (R.T.)

**Keywords:** RNA extraction, protocol, COVID-19, SARS-CoV-2, swabs, qRT-PCR

## Abstract

Background: Rapid diagnosis of COVID-19 is essential in order to restrict the spread of the pandemic, and different approaches for SARS-CoV-2 testing have been proposed as cost-effective and less time-consuming alternatives. For virus detection, the real-time reverse transcriptase–polymerase chain reaction (RT-PCR) technique is still the “gold standard” for accuracy and reliability, but its performance is affected by the efficiency of nucleic acid extraction methods. Objective: In order to improve the SARS-CoV-2 diagnostic workflow, we compared a “standard” commercially available kit, based on viral RNA extraction from human swab samples by magnetic beads, with its technological evolution. The two methods differ mainly in their time consumption (9 vs. 35 min). Methods: We adopted the MAGABIO PLUS VIRUS DNA/RNA PURIFICATION KIT II (BIOER), defined as “standard”, with the automatic extractor BIOER (GenePure Pro fully automatic nucleic acid purification system) to isolate RNA from nasopharyngeal swabs for the detection of SARS-CoV-2 by RT-PCR. We tested this kit with a new faster version of the first one, defined as “rapid” (MAGABIO PLUS VIRUS RNA PURIFICATION KIT II). Results and Conclusion: The two evaluated procedures provided similar analytical results, but the faster method proved to be a more suitable tool for the detection of SARS-CoV-2 from nasopharyngeal swabs, due to a more rapid availability of results, which may contribute to improving both clinical decision making and patient safety.

## 1. Introduction

SARS-CoV-2 was first detected in Wuhan, China [1], as the causative agent of an acute respiratory distress syndrome named Coronavirus Disease 2019 (COVID-19), which still represents the most serious problem for global health. The Chinese government applied severe quarantine measures around the area, but beta-Coronavirus quickly spread over 223 countries with more than 281 million cases [2] and the World Health Organization declared COVID-19 as a pandemic condition on 11 March 2020. Thus far, COVID-19 has induced the need for fast and reliable diagnostic modalities for taking rapid public health actions. Although several efforts have been focused on the development of novel accurate diagnostic tests, to date, real-time reverse transcriptase–polymerase chain reaction (rRT-PCR) assay based on respiratory specimens is still considered the “gold standard” to detect SARS-CoV-2 (WHO, 2020a). In reality, rRT-PCR is a highly sensitive and specific technique, is significantly faster and has a lower potential for contamination and/or errors when compared to other available viral detection methods (e.g., viral antigen detection, standard plaque assay, serology, and electron microscopy), but efficient viral RNA extraction is a prerequisite for the downstream performance of rRT-PCR assays.

In this work, for SARS-CoV-2 detection in nasopharyngeal swabs, we quantified two genome portions that include the genes ORF8 and RNA-dependent RNA polymerase (RdRp). ORF8 codifies for a fast-evolving accessory protein that interferes with the immune response [3]. Therefore, in the early stages of SARS-CoV-2 infection, SARS-CoV-2 ORF8, ORF6, N, and ORF3b are potent interferon antagonists that may delay the release of IFNs, hindering the host’s antiviral response and then benefiting virus replication [4,5]. SARS-CoV-2 ORF8 is a protein composed of 121 amino acids and consists of an N-terminal signal sequence stuck to a predicted Ig-like fold [6]. Within the lumen of the ER, SARS-CoV-2 ORF8 interacts with a variety of host proteins, including many factors involved in ER-associated degradation [7]. The SARS-CoV-2 strains with the ORF8 deletion discovered in different regions of the world [8,9] are considered to still be evolving and facilitate the successful adaption of the virus to different hosts [8,9]. ORF8 is discharged because antibodies are one of the main principal markers of SARS-CoV-2 infection [10]. Several functions have been proposed for SARS-CoV-2 ORF8. ORF8 disturbs IFN-I signaling when it is unfamiliarly overexpressed in cells [11]. It has been shown that ORF8 of SARS-CoV-2, but not ORF8 or ORF8a/ORF8b of SARS-CoV-2, downregulates MHC-I in cells [12].

RdRp is one of the most versatile enzymes of RNA viruses that is indispensable for replicating the genome, as well as for carrying out transcription [13]; it is codified by all ssRNA viruses except for retroviruses [14]. As RdRp of the viruses in this family and others play the main role in infection, it has become the favorite therapeutic target for developing antiviral agents against them [14].

The first step of analysis is crucial for COVID-19 RNA detection, and consists of RNA extraction. Thus far, in these last two years, a growing demand for reliable commercial kits aimed at automatic extraction of SARS-CoV-2 RNA has been observed [15]. The quality and quantity of RNA are important factors because they ensure the accuracy of gene expression analysis and other RNA-based downstream applications [16]. The RNA extraction kit has a crucial responsibility in terms of obtaining high-quality RNA.

Ultimately, other extraction methods have magnetic silica particles that catch nucleic acid, e.g., the NucliSENS easyMAG platform (bioMérieux, Marcy-l’Étoile, France) [17].

We first adopted the MAGABIO PLUS VIRUS DNA/RNA PURIFICATION KIT II (BioER), defined as the “standard” method, with the automatic extractor BioER (GenePure Pro fully automatic nucleic acid purification system) to isolate purified RNA from nasopharyngeal swab for SARS-CoV-2 detection by RT-PCR. Next, we tested this kit with a new faster version, defined as “rapid” (MAGABIO PLUS VIRUS RNA PURIFICATION KIT II). A comparison between the two kits was carried out.

The “standard” kit is certificated for diagnostic use, and is routinely used in diagnostic processes. To achieve authentic or real “standard” positivity results, we used external quality assessment (EAQ). This allowed us to compare between authentic positive and negative samples.

## 2. Materials and Methods

### 2.1. Samples and RNA Extraction

In this work, we investigated a heterogeneous cohort of subjects, comprising symptomatic and non-symptomatic patients hospitalized at the IRCCS Mondino during the pandemic, as well as subjects not hospitalized who accessed the SARS-CoV-22 outpatient clinic. 

RNA from 634 nasopharyngeal swabs collected in universal transport medium (UTM) were extracted with the automated benchtop nucleic acid extraction system NUCLISENS^®^ EASYMAG^®^ (Biomérieux, IT) according to the manufacturer’s specifications. In terms of preparation, the only difference between the two kits is that for the standard kit, it is necessary to add proteinase K (PK) reagent.

RNA from all samples was extracted by two different kits, defined as “standard” and “rapid” (Table 1).

In Table 2, the difference in terms of time consumption for both kits is reported.

In our study, two nucleic acid extraction kits were compared for SARS-CoV2 detection using nasopharyngeal swabs by RT-qPCR analysis, namely, the standard kit (35′) and the rapid kit (9′). The extraction methods used for the SARS-CoV2 detection tests were the same with both kits (magnetic beads), as well as PCR machines and RT-qPCR protocols, but the different chemistry of the rapid RNA extraction kit justified a shorter processing time (9′ against 35′ for the standard) and avoided proteinase K (PK) addiction (10 min to sample lysis). As a result, with the rapid kit, the binding time was also strongly reduced (3 min against 10 min for the standard) and the final elution was achieved in half the time (nearly 3 min) after two washing steps (with Wash 1 and 2). Wash 3 was skipped in the rapid kit protocol. Regarding the storage conditions of the materials, the rapid RNA extraction cartridges were smarter than standard ones, suitable for room temperature (temperature range from 8 to 25 °C) with no specific refrigeration systems to be monitored. 

Ultimately, the rapid RNA extraction protocol may reduce time consumption and optimize sample volumes in order to perform reliable mass diagnosis by RT-qPCR, as observed in Table 2.

The concentration and quality of the RNA extracted from the same 66 samples (as a representative group) with both kits were investigated by nanodrop analysis. With this method, we obtained data on RNA concentration and absorbance at 230, 260, and 280 nm; ng/µL quantification. Additionally, the absorbance ratios 260/280 and 260/230 were chosen as the quantity and quality parameters. Moreover, two samples extracted by the rapid kit that showed low and high concentrations were quantified by Tapetation to confirm the RNA concentration (Appendix A).

### 2.2. Quantitative Real-Time PCR 

Real-time PCR was performed using the SARS-CoV-2 ELITe MGB Kit (ELITechGroup S.p.A.). In this assay, we used a standard reaction mix volume of 30 µL (20 µL of CoV-2 PCR Mix, 0.3 µL of RT Enzyme Mix, and 10 μL of extracted RNA). The real-time PCR protocol included 20 min at 45 °C (RETROTRASCRIPTION), 2 min at 95 °C (DENATURATION), 10 s at 95 °C, 30 s at 60 °C, 10 s at 60 °C, and 20 s at 72 °C (AMPLIFICATION AND DETECTION), repeated for 44 cycles. The one-step fluorescence RT-PCR reaction system allowed us to perform retrotranscription of viral RNA directly during qRT-PCR amplification. Two detection channels were used to amplify ORF8 and RdRp, namely, FAM and ROX, while the VIC channel was used for human RNase P as an internal control (IC). The CFX96™ Real-Time System C1000™ Thermal Cycle (Bio-Rad Laboratories, Inc., Hercules, CA, USA) was used to perform this analysis.

### 2.3. Statistical Analysis

Data are expressed as the mean value ± standard deviation. Statistical differences between means were evaluated using Student’s *t*-test. Statistical analysis was carried out with GraphPad Prism (GraphPad Software Inc., San Diego, CA, USA).

## 3. Results

### 3.1. Comparison between Standard and Rapid Extraction by Nanodrop Analysis

Figure 1A,B show the average of the concentration (in ng/µL) of a group of 66 samples extracted with both kits. In Figure 1A, the extraction by the standard kit showed an average of 13.8 ng/µL, while it was 24.247 ng/µL with the rapid kit. The RNA quantification data showed that extraction by the rapid kit is more efficient in terms of concentration. Figure 1B shows that the two methods of extraction achieved a similar distribution.

Figure 2 shows that the average of the A260/280 RNA isolated using the standard kit was slightly low (1.75), while the samples extracted by the rapid kit had a 260/280 ratio of 1.1. The difference in terms of the 260/280 ratio between the two methods was statistically significant (*p* < 0.001).

In Figure 3, we compared the 260/230 ratio and, with both kits, it was very low. The difference in terms of the 260/280 ratio between the two methods was statistically significant (*p* > 0.001).

### 3.2. Comparison between qPCR Data from Different RNA Extraction Kits

A group of 634 samples were analyzed by qPCR after extraction with both kits. The qPCR protocol measures the presence of IC and two SARS-CoV-2 genes, ORF8 and RdRp. Regarding IC, Table 3 shows that the data obtained from both kits are comparable (Table 3).

In terms of SARS-CoV-2 detection, the SARS-CoV-2 gene quantification by qPCR showed some differences between the samples extracted by the different kits.

Table 4 provides a summary of the results based on the qPCR kit datasheet that considers SARS-CoV-2 samples to be positive (one of the two genes ORF8 or RdRp).

Figure 4 shows an example of a positive sample for both kits, and the curves are stackable.

Fifty-one samples showed a discrepancy between the two kits; 43 were positive for SARS-CoV-2 only after rapid extraction, while eight were detectable only with standard extraction. 

Even if we considered all samples positive with at least one COVID-19 gene detection, the FDA guidelines (https://www.fda.gov/media/134922/download, accessed on 7 January 2021) suggest that detection beyond 40 Ct may provide a false-positive. Interestingly, 53% (23/43) of the samples that detected a COVID-19-positive result only with the rapid kit showed a Ct < 40, mainly between 35 and 38 Ct, while the few samples (8) detected only with the standard kit showed a low amount of SARS-CoV-2 RNA, with Ct > 40 (Figure 5).

## 4. Discussion 

This manuscript described the evaluation of two commercially available RNA extraction kits that mainly differ in terms of time consumption (9 vs. 35 min). More specifically, a new, faster technique of nucleic acid extraction was compared to the “reference” method, which represents the evolution. As already reported in the literature, the impact of RNA extraction is fundamental in SARS-CoV-2 detection [18,19] and, on the contrary, in diagnostics, the data about time consumption are also very important [20]. In fact, other papers have already demonstrated that performing a combination of rapid and quality RNA isolation is possible [20].

Komiazyk et al. showed that the lower the viral load, the more difficult it is to interpret the results. This appears to be due to problems with available diagnostic tests failing this low viral load, which is still capable of infection [18]. With CT ≥ 35, the test also reacts with some portion of the virus, while the virus itself is no longer present; therefore, the subject is no longer contagious.

The first data concerning RNA quantification showed that extraction by the rapid kit is more efficient in terms of the concentration. In fact, in ng/µL, the concentration obtained by the rapid kit was two times that of the standard kit quantification. Regarding RNA quality, the standard kit results showed good ratios of 260/280 and 260/230, while the data obtained by the rapid kit showed a low absorbance ratio. Usually, low 260/280 and 230/260 ratios are caused by residual phenols or other reagents associated with the extraction protocol, and we hypothesized that it may be caused by the short wash steps of the rapid kit. 

On the contrary, the sensibility of the kits in terms of SARS-CoV-2 detection seems to be increased using the rapid kit; in fact, the percentage of positive samples was 21.1%, and while using the standard kit, we detected 15.6% of the samples as positive for SARS-CoV-2. These data suggest that the low 260/280 and 260/230 ratios do not interfere in viral RNA detection.

These data suggest that this new rapid kit has been optimized in the binding, elution, and lysis steps, making PK no longer required. Once we have confirmed the accuracy and reliability of this new method, it could be introduced into daily routine to reduce the time-processing of nasopharyngeal swabs for SARS-CoV-2 detection, thus making the whole analytical process more efficient and shortening diagnostic procedures. 

This new kit demonstrated suitability for the extraction of viral RNA from serum, plasma, tissue, swabs (after sampling), stools, and body fluid samples, completing the nucleic acid extraction process in only 9 min. Reliable and strict, this new application can be suggested to be a precious tool for the detection of SARS-CoV-2 from nasopharyngeal swabs in a pandemic context, where it is relevant to obtain rapid and accurate results to diagnose patients and employ rapid public health actions. As already published, the setting up of rapid nucleic acid extraction procedures may be very useful for reliable diagnosis [20]. In reality, this faster method for RNA extraction (a) reduces the turnaround time, (b) improves outcomes and workflow, and (c) accelerates clinical decision making.

Furthermore, even once the COVID-19 pandemic is over, this fast technique of nucleic acid extraction could increase the efficiency of the diagnostic process regarding other infectious diseases of clinical impact on public health.

## Figures and Tables

**Figure 1 diagnostics-12-01561-f001:**
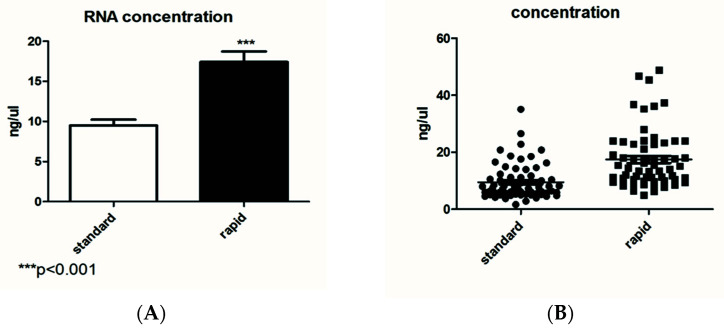
Comparison between the RNA concentration obtained by the standard and rapid extraction kits. (**A**) Statistical difference (*** *p* < 0.001); (**B**) the distribution of the analyzed samples (*N* = 66 for each group).

**Figure 2 diagnostics-12-01561-f002:**
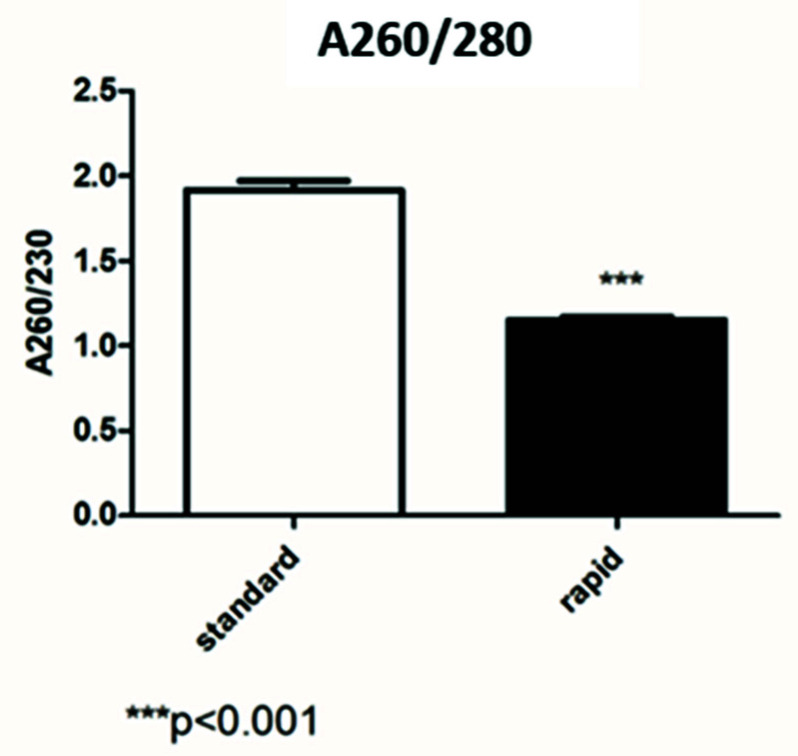
Comparison between the 260/280 ratio obtained by the standard and rapid extraction kits.

**Figure 3 diagnostics-12-01561-f003:**
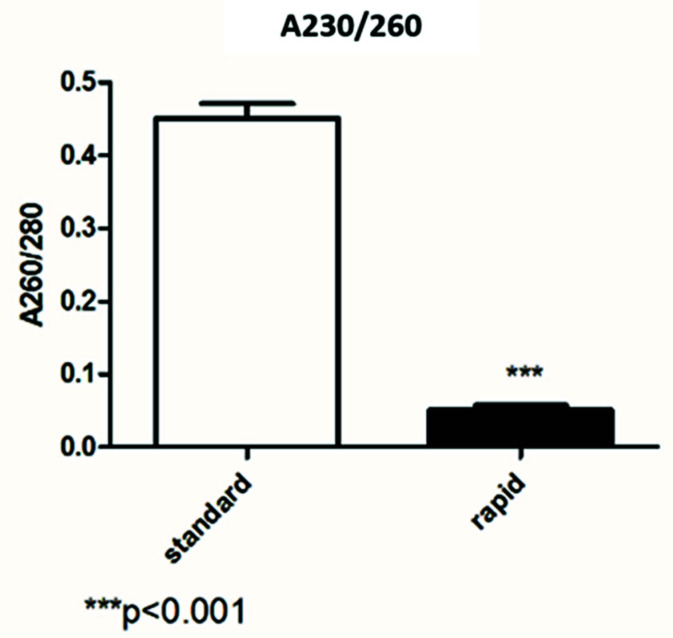
Comparison between the 230/260 ratio obtained by the standard and rapid extraction kits.

**Figure 4 diagnostics-12-01561-f004:**
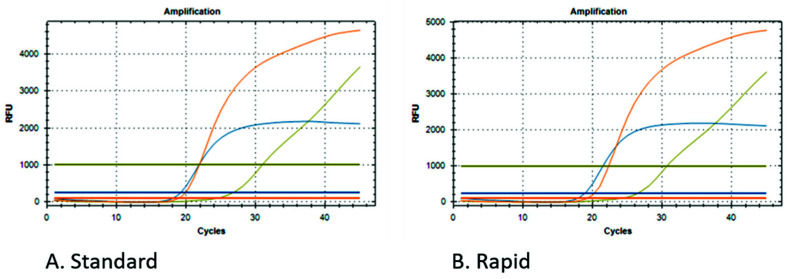
Examples of the qPCR curves obtained from the analysis of the same samples after extraction with the standard (**A**) and rapid (**B**) kits.

**Figure 5 diagnostics-12-01561-f005:**
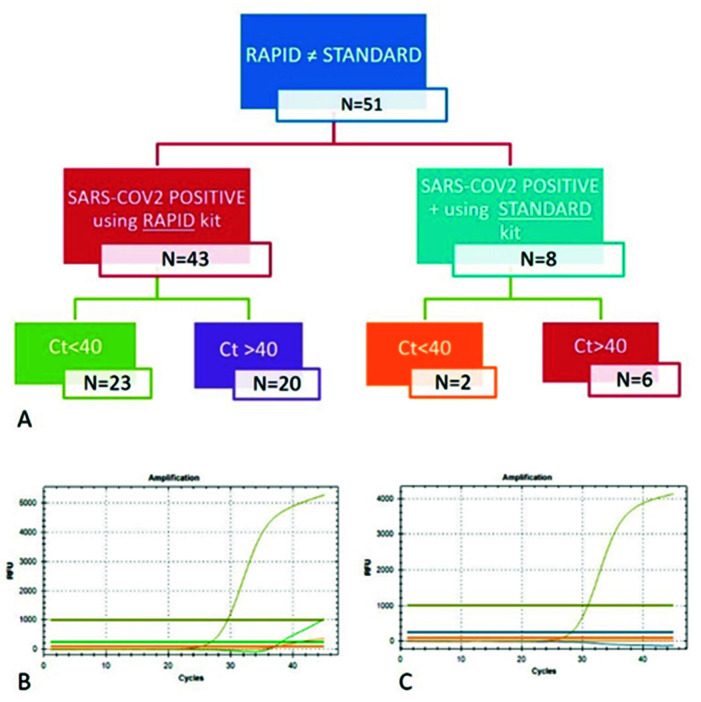
Schematic representation of the samples that showed divergence in SARS-CoV-2 detection (**A**). Examples of the qPCR curves obtained for the analysis of the same samples after extraction with the rapid (**B**) and standard (**C**) kits that showed different results.

**Table 1 diagnostics-12-01561-t001:** Schematic representation of the time consumption in each step for the standard and rapid kit protocols.

			Time	
			Standard	Fast
Step	Well	Name		
1	1	lysis	10:00:00 (mixing)	no
2	6	beads	00:45 (mixing and magnet)	00:15 (magnet)
3	1	binding	10:35 (mixing and magnet)	03:35 (mixing and magnet)
4	2	wash 1	02:30 (mixing and magnet)	00:50 (mixing and magnet)
5	3	wash 2	01:30 (mixing and magnet)	00:50 (mixing and magnet)
6	4	wash 3	01:30 (mixing and magnet)	no
7	5	elution	07:35 (waiting, mixing and magnet)	03:25 (waiting, mixing and magnet)
8	6	discard	00:30 (mixing and magnet)	00:00
**total time**			**35 min**	**9 min**

**Table 2 diagnostics-12-01561-t002:** Summary of the comparison between the rapid and standard kits.

Protocols	Standard	Rapid
*Methods*	Magnetic beads	Magnetic beads
*Sample volume*	20–1000 μL	10–300 μL
*Elution volume*	80 μL	70 μL
*Processing time*	35 min	9 min
*Proteinase K*	Yes	No

**Table 3 diagnostics-12-01561-t003:** Comparison of IC detection, by qPCR, between 634 samples extracted in parallel with the standard and rapid kits. The Ct average and standard deviation parameters are reported.

	IC Ct	
Kit	*Average*	*Standard Deviation*
*Standard*	28.79	1.48
*Rapid*	28.5	1.53

**Table 4 diagnostics-12-01561-t004:** Summary of SARS-CoV-2 detection by qPCR using the standard and rapid kits for RNA isolation.

*STANDARD*		*RAPID*	
POS	NEG	POS	NEG
99	535	134	500

## Data Availability

Datasets for this manuscript are publicly available, as they are linked to 10.5281/zenodo.6477268, accessed on 1 May 2022.

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
