# Peer review of "Comparison of Two RNA Extraction Methods for the Molecular Detection of SARS-CoV-2 from Nasopharyngeal Swab Samples"

_diagnostics, 2022, doi:10.3390/diagnostics12071561_

Round 1

Reviewer 1 Report

Major revision is required before the acceptance of this manuscript in diagnostics. Some of the specific issues that support this recommendation are:

  1. Based on the A230/260 ratio results, it seems that RNA isolated from both approaches have very low ratios, which can be caused by residual contamination from the isolation process. Considering that RNA concentration (Figure 1) is determined by absorbance measurement using Nanodrop, the concentration results could also be affected by the low A230/260 ratio. Therefore, the manuscript needs to either demonstrates the accuracy of concentration measurements with Nanodrop (considering such low A230/260 ratio), or measure RNA concentration using a different method, such as intercalating dye.
  2. Is there any authentic or real “standard” positivity results of the 618 samples used in the manuscript. Considering the fact both RNA extraction approaches described in the manuscript suffer from residual contamination, the accuracy (the number of positive/negative samples) of these two approaches should be compared with authentic positive/negative samples, instead of simply compare with each other.
  3. What are the 66 samples described in section 3.1? Are they included in the 618 swab samples?
  4. In the Materials and Methods section 2.1, it described that “300uL samples” were loaded on the 96-well plate in both approaches. Where does the 300uL samples come from?

Author Response

Reviewer 1

Comments and Suggestions for Authors

Major revision is required before the acceptance of this manuscript in diagnostics. Some of the specific issues that support this recommendation are:

  1. Based on the A230/260 ratio results, it seems that RNA isolated from both approaches have very low ratios, which can be caused by residual contamination from the isolation process. Considering that RNA concentration (Figure 1) is determined by absorbance measurement using Nanodrop, the concentration results could also be affected by the low A230/260 ratio. Therefore, the manuscript needs to either demonstrates the accuracy of concentration measurements with Nanodrop (considering such low A230/260 ratio), or measure RNA concentration using a different method, such as intercalating dye.

We thank the reviewer for all the suggestions.  Unfortunately, the analyzed RNA have been used for other projects and the amount of material is very low. For this reason, we have also measured RNA concentration by Agilent TapeStation system only for two samples.

Both samples were extracted by rapid kit that showed low A230/260 ratio, the concentration of these samples has been confermed by TapeStation Analysis (supplementary fugure S1). All nanodrop quantification are in the supplementary table 1 and 2 (S1 and S2).

  1. Is there any authentic or real “standard” positivity results of the 618 samples used in the manuscript. Considering the fact both RNA extraction approaches described in the manuscript suffer from residual contamination, the accuracy (the number of positive/negative samples) of these two approaches should be compared with authentic positive/negative samples, instead of simply compare with each other.

Thanks for this question. The “standard” kit that we use to compare the “rapid” kit is routinely used in diagnostic process so we assume that the residual contamination do to alter the results.

In fact the “standard” kit is certificate for diagnostic use and it is also use, with successful results, for External quality assessment.

  1. What are the 66 samples described in section 3.1? Are they included in the 618 swab samples?

Thanks for this puntualization, yes, 66 samples in section 3.1 are part of total 618 swab samples.

  1. In the Materials and Methods section 2.1, it described that “300uL samples” were loaded on the 96-well plate in both approaches. Where does the 300uL samples come from?

Thanks for this question, for 300 ul are from UTM (Universal Transport Medium) where are allocated nasal flocked swabs.

Reviewer 2 Report

The article describes comparison of efficiency of two RNA extraction methods for SARS-CoV-2 detection by RT-PCR. The manuscript seems to be very useful for diagnosticians and scientists working to improve virus detection methods.

My overall impression is that the article is written in an inaccurate and chaotic way and needs to be thoroughly improved and filled with missing content. Many issues should be cosider/ improve by the authors of manuscript before publishing it. Below are my suggestions in points:

  1. Introduction: In my opinion line 86-92 is not necessary, contributes nothing important to the whole story.
  2. M&M: please explain the abreviations UTM
  3. M&M: Could the authors write something more about the tested samples? where did they come from? who were the patients? or did they have symptoms? as part of what was sampled?
  4. M&M: section 2.1.1 and 2.1.2 where the authors provide protocol (?) is not necessary. This part confuses the reader, especially that in line 106-107 the authors rovide information about tested extraction methods.
  5. Results: It’s a pity that only 66 samples where compared after RNA isolation with nanodrop. analysis In my opinion is very poor numer, especially that the authors wrote that they collected 618. Could the authors include the other measurement results for the remaining samples to the results?
  6. Results: line 179: Please give more details about the used „nanodrop analysis”
  7. Results: the resutls which are shown in Figure 1A and B, Figure 2 and Figure 3 should be described in the text and commented on, in the results section.
  8. Results: The authors should prepared all results in table with Ct of real time PCR for 614 samples for both extraction methods in Supplemenary materials.
  9. Results: Figure 4: please change „fast”for „rapid”
  10. Disscusion: is too short!!! The authors should include here other studies with extraction kits used for koronaviruses diagnosis. Furthermore, please include here section with explanation of idea „rapid” isolation method.
  11. Please formulate your conclusions.
  12. The edition English service is needed to improve this article.

Author Response

Reviewer 2

The article describes comparison of efficiency of two RNA extraction methods for SARS-CoV-2 detection by RT-PCR. The manuscript seems to be very useful for diagnosticians and scientists working to improve virus detection methods.

My overall impression is that the article is written in an inaccurate and chaotic way and needs to be thoroughly improved and filled with missing content. Many issues should be cosider/ improve by the authors of manuscript before publishing it.

Below are my suggestions in points:

  1. Introduction: In my opinion line 86-92 is not necessary, contributes nothing important to the whole story.

Thanks for this suggestion, we have cut this this part.

  1. M&M: please explain the abreviations UTM

Yes, sorry for this missing information, we have added in the manuscript

  1. M&M: Could the authors write something more about the tested samples? where did they come from? who were the patients? or did they have symptoms? as part of what was sampled?

Thanks for this question, we added in the text

“In this work we have investigated an heterogeneous cohort of subjects that included symptomatic and non-symptomatic patients hospitalized at the IRCCS Mondino during the pandemic period and also subjects not hospitalized that access to the SARS-CoV-22 outpatient clinic.”

  1. M&M: section 2.1.1 and 2.1.2 where the authors provide protocol (?) is not necessary. This part confuses the reader, especially that in line 106-107 the authors rovide information about tested extraction methods.

We are agree, we removed sections 2.1.1 and 2.1.2, and the protocols are explained in table 1.

  1. Results: It’s a pity that only 66 samples where compared after RNA isolation with nanodrop. analysis In my opinion is very poor numer, especially that the authors wrote that they collected 618. Could the authors include the other measurement results for the remaining samples to the results?

Unfortunately, we can not quantify more samples because no biological materials is still available. On the other hand our data (figure 1) showed that the analyzed samples are homogenous in terms of ng quantification.

  1. Results: line 179: Please give more details about the used „nanodrop analysis”

Thanks, done.

  1. Results: the resutls which are shown in Figure 1A and B, Figure 2 and Figure 3 should be described in the text and commented on, in the results section.

Thanks, we have added more comments about these figures.

  1. Results: The authors should prepared all results in table with Ct of real time PCR for 614 samples for both extraction methods in Supplemenary materials.

Thank you for this suggestion, the requested file has been prepared as supplementary (S3)..

We are sorry but there was a mistake in the text, the analyzed samples by qPCR are 634.

  1. Results: Figure 4: please change „fast”for „rapid”

Thanks, we have changed figure 4.

  1. Discussion: is too short!!! The authors should include here other studies with extraction kits used for koronaviruses diagnosis. Furthermore, please include here section with explanation of idea „rapid” isolation method.

Thank you, we are agree, the discussion is too brief, we have improved this part

11. Please formulate your conclusions.

Yes, we have added some sentence for conclusions.

  1. The edition English service is needed to improve this article.

Thanks for your kind suggestion, we have submitted our work to MDPI English editing

Round 2

Reviewer 1 Report

The authors have addressed all the questions from last review.

Reviewer 2 Report

Dear Authors,

Thank you very much for consider my suggestion.

I am satisfied with corrections which you made.  In my opinion the article should be publish.